# Thyroid Transcriptomic Profiling Reveals the Follicular Phase Differential Regulation of lncRNA and mRNA Related to Prolificacy in Small Tail Han Sheep with Two *FecB* Genotypes

**DOI:** 10.3390/genes13050849

**Published:** 2022-05-10

**Authors:** Cheng Chang, Xiaoyun He, Ran Di, Xiangyu Wang, Miaoceng Han, Chen Liang, Mingxing Chu

**Affiliations:** 1Key Laboratory of Animal Genetics, Breeding and Reproduction of Ministry of Agriculture and Rural Affairs, Institute of Animal Sciences, Chinese Academy of Agricultural Sciences, Beijing 100193, China; changcheng20200911@163.com (C.C.); hedayun@sina.cn (X.H.); diran@caas.cn (R.D.); wangxiangyu@caas.cn (X.W.); 2College of Animal Science, Shanxi Agricultural University, Jinzhong 030801, China; hanmiaoceng@163.com

**Keywords:** lncRNAs, RNA-Seq, thyroid, prolificacy, sheep

## Abstract

Long non-coding RNA (lncRNA) accounts for a large proportion of RNA in animals. The thyroid gland has been established as an important gland involved in animal reproduction, however, little is known of its gene expression patterns and potential roles in the sheep. Herein, RNA-Seq was used to detect reproduction-related differentially expressed lncRNAs (DELs) and mRNAs (DEGs) in the follicular phase (FT) *FecB^BB^* (MM) and *FecB^++^* (ww) genotypes of Small Tail Han (STH) sheep thyroids. Overall, 29 DELs and 448 DEGs in thyroid between MM and ww sheep were screened. Moreover, GO and KEGG enrichment analysis showed that targets of DELs and DEGs were annotated in biological transitions, such as cell cycle, oocyte meiosis and methylation, which in turn affect reproductive performance in sheep. In addition, we constructed co-expression and networks of lncRNAs-mRNAs. Specifically, *XLOC*_*075176* targeted *MYB*, *XLOC*_*014695* targeted *VCAN*, *106991527* targeted *CASR*, *XLOC*_*075176* targeted *KIFC1*, *XLOC*_*360232* targeted *BRCA2*. All these differential lncRNAs and mRNAs expression profiles in the thyroid provide a new resource for elucidating the regulatory mechanism underlying STH sheep prolificacy.

## 1. Introduction

Small Tail Han (STH) sheep is an excellent breed in China. It has the characteristics of year-round estrous, strong fecundity, stable performance heredity, and the average lambing rate is more than 250% [1]. *FecB* gene also known as the bone morphogenetic protein receptor 1B (*BMPR1B*) gene, was first identified as the main gene for high fecundity in sheep due to the multiple lambing phenomenon caused by *FecB* mutation [2]. *FecB* gene was located on chromosome 6 of sheep, and there were three genotypes: *FecB^BB^* (double copy *FecB* mutation), *FecB^B^*^+^ (single copy *FecB* mutation), and *FecB^++^* (no copy *FecB* mutation) [3]. As a receptor for bone morphogenetic proteins (BMP), it is also involved in the regulation of follicle-stimulating hormone (FSH) levels, thereby regulating ovulation rates in animals and humans [4]. On days 9–11 of estrus, the progesterone concentration of *FecB^BB^* genotype Booroola Merino sheep was 25% higher than that of *FecB^++^* Merino sheep [5]. The ovulation number of ewes with *FecB^BB^* genotype was significantly higher than that of the other two genotypes, *FecB* mutations inhibit granulosa cell apoptosis in sheep, prevent follicular atresia, promote ovulation and increase litter size. This may be an important physiological mechanism by which the *FecB* gene affects sheep fertility [6]. According to the previous research report of our group, all three genotypes of *FecB* are distributed in STH sheep, and these three genotypes are significantly correlated with the litter size of ewes [7].

Reproduction is one of the main characteristics that affect the economic benefits of sheep and increasing the number of lambs is an effective way to improve the economic benefits of sheep breeding [8]. LncRNAs are a kind of RNA with a length of more than 200 nucleotides and no open reading frame and protein-coding function. It exists in many mammalian tissues and is located in the nucleus and cytoplasm. It plays an important role in biological processes, such as differentiation, signal transduction, and immune regulation [9]. Hayashizaki et al. [10] showed that in mammals, only 2% of RNA is protein-coding, and more than 70% is the LncRNA. Through QTL analysis, Li et al. [11] found that 127 DELs could regulate the number of corpus luteum, litter size, androsterone, live-born litter, stillbirth, follicle-stimulating hormone concentration, and viable embryos. Using RNA sequencing technology, He et al. [12] found that there were 346 DELs and 186 DEGs in the distal pituitary of sheep with short and long photoperiods. Liu et al. [13] identified 140 DELs and 1554 DEGs at different stages of porcine ovary development. These LncRNAs may affect the animal reproductive process through the *PI3K-Akt* signaling pathway [14], *TGF-β* signaling pathway, *MAPK* signaling pathway and *Wnt* signaling pathway [15].

The thyroid gland is important in mammals. It functions through the hypothalamus-pituitary-thyroid axis and can regulate various metabolic activities of the body. The thyroid gland mainly secretes thyroid hormones, which can regulate mitochondrial oxygen consumption, carbohydrates, and fats. Metabolic and cardiovascular development regulated fetal growth and development [16]. Feng et al. [17] found that T3 affects eel gonadal differentiation genes and may play a role in regulating eel male development and sex reversal. Costa et al. [18] added T3 to the in vitro maturation medium of bovine oocytes, which had a certain promoting effect on embryonic development. Song et al. [19] showed that T4 affected the accumulation of porcine 17β-estradiol and induced apoptosis, but not the proliferation of granulosa cells.

In light of these discoveries, the study presented here was focused on analyzing transcriptomic differences in the thyroid of STH sheep in the follicular phase (FT) between the MM group and ww group, to detect the DELs and DEGs, and to predict their potential function that related to sheep reproduction. This investigation is essential for better understanding the molecular mechanisms of lncRNAs in regulating sheep reproduction of different genotypes, and also will provide insight into other female mammals.

## 2. Materials and Methods

### 2.1. Animal Tissue Collection

STH sheep were obtained from a core breeding flock of sheep in the Lucian region of Shandong, China, and healthy infertile sheep aged 2–4 years were adopted from the jugular vein (*n* = 890) and their *FecB* genotype was identified using the TaqMan probe [20], then six sheep (3MM and 3ww, respectively) of comparable size and condition were selected for the study [3]. All sheep were treated with the Controlled Internal Drug Releasing device (CIDR) for synchronized estrus. Six sheep were euthanized 50 h after synchronized estrus treatment and thyroid tissue collected was stored temporarily in liquid nitrogen and brought back to the laboratory for immediate storage in a −80 °C refrigerator for later experimental use.

### 2.2. Total RNA Extraction and Library Construction

The thyroid gland was extracted from six STH sheep and total RNA was extracted using TRIzol reagent (Carlsbad, CA, USA). Agarose gel electrophoresis was used to detect whether the RNA was contaminated and the RNA was tested for purity and concentration using the RNA Nano 6000 Assay Kit of the Bioanalyzer 2100 system (Technologies, CA, USA), Nano Photometer^®^ spectrophotometer and Qubit^®^ RNA Assay Kit in Qubit^®^ 2.0 Fluorometer (Technologies, San Jose, CA, USA) for RNA purity and concentration. The rRNA was removed using the Epicentre Ribo-zero™ rRNA Removal Kit and the NEB Next^®^ Ultra™ Directional RNA Library Prep Kit for Illumina^®^ (NEB, Ipswich, MA, USA) was used to construct the RNA library, and the Illumina Hiseq 4000 platform (Illumina) was used for sequencing [9].

### 2.3. Processing of Sequencing Data

We filtered the raw sequencing data to remove low-quality reads to obtain clean reads and calculate the Q30 and GC content of clean reads. HISAT2 v. 2.0.4 was used to align paired-end clean reads of each sample to sheep reference genome Oar v. 4.0. The mapped reads of each sample were assembled by StringTie v. 1.3.1 [21].

### 2.4. Identification of Candidate lncRNAs

Transcripts are identified by the following process: (1) Use Cuffmerge to remove transcripts of uncertain orientation. (2) Select transcripts with exon number ≥2 and length >200 nt. (3) Use Cuffdiff v. 2.1.1 to calculate the length of each transcript and select transcripts with FPKM ≥ 0.5. (4) The CPC v. 2.81 [22], CNCI v. 2.0 [23] and PFAM v. 1.3 [24] tools were selected to measure the coding capacity of the proteins.

### 2.5. Analysis of Differentially Expressed Genes

FPKM values were used to assess the expression levels of transcripts [25] and experiments included three biological replicates using the DEseq2 package of R software to identify differentially expressed transcripts [26]. LncRNAs and mRNAs with a *p* < 0.001 were considered to be differentially expressed genes between the two groups. The number of differential genes up-or down-regulated was mapped using Prism software (version 8.3.0), and the Wayne map was created using the website Venny2.1.0 (https://bioinfogp.cnb.csic.es/tools/venny/index.html (accessed on 22 January 2022), and the heat map was created using the R software Pheatmap package.

### 2.6. Bioinformatics Analysis

Enrichment analysis using Gene Ontology (GO) and the Kyoto Encyclopedia of Genes and Genomes (KEGG). Enrichment analysis was performed on screened DELs_targets and DEGs using the Metascape (https://metascape.org/gp/index.html#/main/step1, accessed on 20 April 2022) database, and *p* < 0.001 were considered significant differences for GO term and KEGG passway. Based on significant differences, the top 15 terms for GO and the top 20 pathways for KEGG were selected for display in the text (DEGs were enriched to a total of six pathways, all of which are displayed in the text, all with *p* < 0.001). Horizontal bar charts and enrichment bubble diagrams were produced using the Bioinformatics (http://www.bioinformatics.com.cn/, accessed on 20 April 2022).

### 2.7. Construction of a Network for the Analysis of lncRNAs-mRNAs Interactions

The intersection of DELs_targets and DEGs was chosen to construct the reciprocal network, and Cytoscape (version 3.7.1) was chosen for the software [27].

### 2.8. Reverse Transcription and qPCR Validation

For the qPCR analysis of mRNAs and lncRNAs, reverse transcription was performed using the PrimeScript^TM^ RT reagent kit (TaKaRa, Dalian, China). Furthermore, qPCR with the SYBR Green qPCR Mix (TaKaRa, Dalian, China) was conducted with a RocheLight Cycler R 480 II system (Roche Applied Science, Mannheim, Germany) as follows: initial denaturation at 95 °C for 5 min, followed by 40 cycles of denaturation at 95 °C for 5 s, then annealing at 60 °C for 30 s. qPCR and RNA-seq mapping using Prism software (version 8.3.0).

## 3. Results

### 3.1. Summary of Raw Sequence Reads

After removing low-quality sequences, a total of 278,914,864 and 269,240,138 clean reads with greater than 92.46% of Q30 were obtained in MM_FT and ww_FT. Approximate 89.46% to 90.11% of the reads were successfully mapped to the *Ovis aries* reference genome (Table 1).

### 3.2. Differential Expression Analysis of lncRNAs and mRNAs

A total of 14,548 lncRNAs (including 13,023 known lncRNAs and 1525 novel lncRNAs) and 33,904 mRNAs were identified from MM_FT and ww_FT. There were 13 DELs and 237 DEGs were upregulated, 16 DELs and 211 DEGs were downregulated (Figure 1A,B), and the DELs (Figure 1C) and DEGs (Figure 1D) showed different expression patterns between MM_FT and ww_FT. All DELs (*p* < 0.001) and DEGs (*p* < 0.001) were statistically significant. Venn diagram visually showed the numbers of common and unique DELs_targets and DEGs among MM_FT and ww_FT (Appendix A).

### 3.3. GO Analysis of the Biological Function of DELs and DEGs

GO annotation enrichment analysis is an internationally recognized and classified standard system for gene function and is often used to describe functions of the DELs and DEGs involved in cellular components, molecular functions, and biological processes. The figure shows the first 15 pathways enriched by DELs in these three processes (Figure 2A, Appendix A). The targeted genes in pathways enriched by DELs were involved in biological processes which were related to lipid transport, immune system process, and immune response, the enrichment of mitotic cell cycle GO term was the most significant, which contains 100 genes. Involved in cellular components which were condensed chromosomes, chromosomes, centromeric region and mitotic spindle and so on, but the most important term is the chromosomal region, which contains 70 genes, and is involved in molecular function including kinase binding, protein kinase binding, and microtubule binding and so on. The most enriching term is tubulin binding, which contains 39 genes.

The first 15 terms enriched by DEGs in the three processes are shown in the figure (Figure 2B, Appendix A). Pathways enriched by DEGs were involved in biological processes related to histone modification, immune system process and immune response, the most important term is microtubule cytoskeleton organization, which contains 19 genes, also in cellular components including cell body, microtubule and nuclear matrix, the enrichment of axon GO term was the most significant, which contains 22 genes, and in molecular function, pathways included core promoter sequence-specific DNA binding and protein-lysine N-methyltransferase activity and so on, but the most enrichment term is tubulin binding, which contains nine genes.

### 3.4. KEGG Pathway Analysis

KEGG is a primary public pathway database to understand the potential function of DEGs. DELs_targeted mRNAs were associated with pathways, such as Oocyte meiosis, cell cycle and the p53 signaling pathway (Figure 3A, Appendix A). DEGs were enriched in the pathways which involved hematopoietic cell lineage, primary immunodeficiency, and dopaminergic synapse (Figure 3B, Appendix A).

### 3.5. Interaction Analysis of lncRNAs-mRNAs and Function Prediction

To better understand the relationship between lncRNAs and mRNAs, after selecting the intersection of DELs_targets and DEGs, we constructed a network of lncRNAs-mRNAs interactions to further understand which EDGs play a role in the regulation of sheep reproduction. Between MM_FT and ww_FT, a total of 29 DELs and 448 DEGs were involved in the network. In the regulatory network, *ICA*, *CRAC1*, *SCN3A* and *NEC2* were simultaneously regulated by *LNC*_*010851* and *XR*_*001435524*.1, and *TAF1* was simultaneously regulated by *LNC*_*000701* and *LNC*_*003851*. Red (Figure 4A) and green (Figure 4B) indicate up- or down-regulation, respectively (Appendix A).

### 3.6. Data Validation

To assess the accuracy of sequencing, mRNAs and lncRNAs were selected randomly for qPCR validation. We measured the gene expression level and compared it with the RNA-seq data. The results demonstrated that RNA-seq data and qPCR data showed similar patterns, which indicates the reliability of the data generated from RNA-seq (Figure 5, Appendix A). The sequences of qPCR primers are listed in Table 2.

## 4. Discussion

The study found that lncRNAs were involved in a variety of reproductive functions, such as spermatogenesis, placenta formation, sex hormone response signaling pathways, and gonadogenesis [28]. In this study, multiple thyroid function-related lncRNAs were identified from MM_FT and ww_FT by RNA-seq technology, and the length and exon number of the detected lncRNAs were also analyzed, and the average length of the discovered lncRNAs was also analyzed. The average length of the newly identified lncRNAs is 1312 nt and the number of exons is 4.5.

In this study, 29 DELs and 448 DEGs were identified in comparing the two groups of MM_FT and ww_FT, respectively. In the functional analysis of differential genes in the hypothalamus during the follicular phase of single-multiple lamb sheep, we found several key genes, including proto-oncogene transcription factor (*MYB*), versican (*VCAN*), DNA repair associated (*BRCA2*), kinesin family member C1 (*KIFC1*) and calcium-sensing receptor (*CASR*).

MYB proteins are key factors in regulatory networks for biotic regulation of development, metabolism, and responses to biotic and abiotic stresses [29]. MYB can play a role as a transcription factor. Zheng et al. [30] established a mouse in vitro culture model and found that *MYB* can regulate mouse oocyte maturation by activating cytokines and genes, such as *MAPK* and *MPF*. Later, the team found [31] that *MYB* can mediate the *MPF* activation pathway to induce mouse oocyte maturation.

Versican (*VCAN*), a chondroitin sulfate proteoglycan, refers to a major part of the extracellular matrix [32]. Shen et al. [33] showed that the expression level of *VCAN* in cumulus cells was positively correlated with the early embryo morphology score. With the deepening of the development of cumulus cells, the expression level of *VCAN* can be used as an indicator to evaluate the developmental ability of oocytes. Özler et al. [34] showed that serum *VCAN* levels in patients with polycystic ovary syndrome (PCOS) were significantly reduced, and there is a positive correlation between VCAN and the expression of ADAMTS-1, indicating that VCAN and ADAMTS-1 may promote each other in the expression process. ADAMTS-1 can affect follicular development, ovulation, luteal formation and degeneration by affecting the extracellular matrix (ECM), indicating that *VCAN* may play a role in ovulatory dysfunction and the pathogenesis of PCOS.

The calcium-sensing receptor (*CASR*) is a G protein-coupled receptor of the C family. Activation of *CASR* leads to intracellular Ca^2+^ mobilization, regulation of intracellular cAMP levels, and activation of multiple protein kinases [35]. Liu et al. [36] found that *CASR* agonists significantly up-regulated the expression of hyaluronan synthase 2 (*HAS2*) in porcine cumulus cells, while CASR inhibitors down-regulated all *HAS2*, prostaglandin-endoperoxide synthase 2 (*PTGS2*) and tumor necrosis. The expression of factor α-inducible protein 6 (*TNFAIP6*) indicates that CASR activity plays an important role in FSH-stimulated porcine cumulus expansion. Yang et al. showed that *CASR* is involved in rat oocyte post-ovulatory aging (POA) and rat oocyte spontaneous activation (AS) by blocking Ca^2+^ channels [37].

The kinesin-like protein *KIFC1* is a nonessential minus end-directed motor of the kinesin-14 family [38]. Mihalas et al. [39] found that using siRNA to interfere with *KIFC1* expression in mouse oocytes reduced oocyte meiosis in healthy young mice (4–6 weeks). The study had shown that treatment of mouse oocytes with bisphenol reduces the expression of *KIFC1* in cells, which in turn impairs oocyte meiosis and genome stability [40].

Multiple studies showed that mutations in the *BRCA2* gene can adversely affect reproduction in animals [41]. Miao et al. [42] knocked out the *BRCA2* gene in mouse oocytes using Gdf9-Cre technology to generate *BRCA2* gene-deficient mice and found that this type of mice was infertile due to the accumulation of DNA damage leading to follicular development arrest and oocyte quality defects. Gad et al. [43] used RNA high-throughput sequencing technology to identify and analyze the transcriptome profile of pig oocytes from large and small follicles and found that *BRCA2* was a significantly differentially expressed gene, proving that the *BRCA2* gene plays an important role in the development of pig oocytes.

Overall, these five genes play an important role in animal reproduction. In addition, these DEGs are also enriched in important pathways, such as cell cycle, p53 signaling pathway and oocyte meiosis, which play a very important role in the biosynthesis of animal reproductive hormones.

## 5. Conclusions

In summary, the thyroid transcriptome study revealed differential expression of lncRNAs and mRNAs related to prolificacy in sheep with different *FecB* genotyping. We screened several sets of target genes of DELs and DEGs under reproductive cycle and genotypes, all of which played an important role in animal metabolism. Additionally, to reveal the role of these DEGs in sheep reproduction, we constructed a network map of lncRNAs-mRNAs interactions, and these DELs and DEGs provide some new insights into the mechanism of thyroid regulation of sheep prolificacy.

## Figures and Tables

**Figure 1 genes-13-00849-f001:**
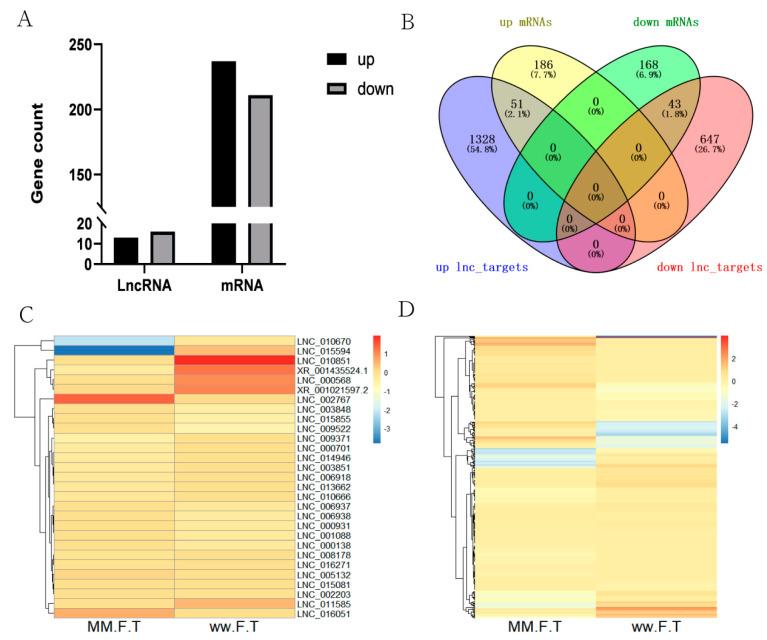
(**A**) Histogram representing the numbers of upregulated and downregulated lncRNAs and mRNAs in sheep thyroid between MM_FT and ww_FT. (**B**) Venn diagram representing the overlapping numbers of DELs_targets and DEGs in MM_FT vs. ww_FT. Heat maps showing the expression intensity of 448 DEGs (**C**) and 29 DELs (**D**) in MM_FT vs. ww_FT.

**Figure 2 genes-13-00849-f002:**
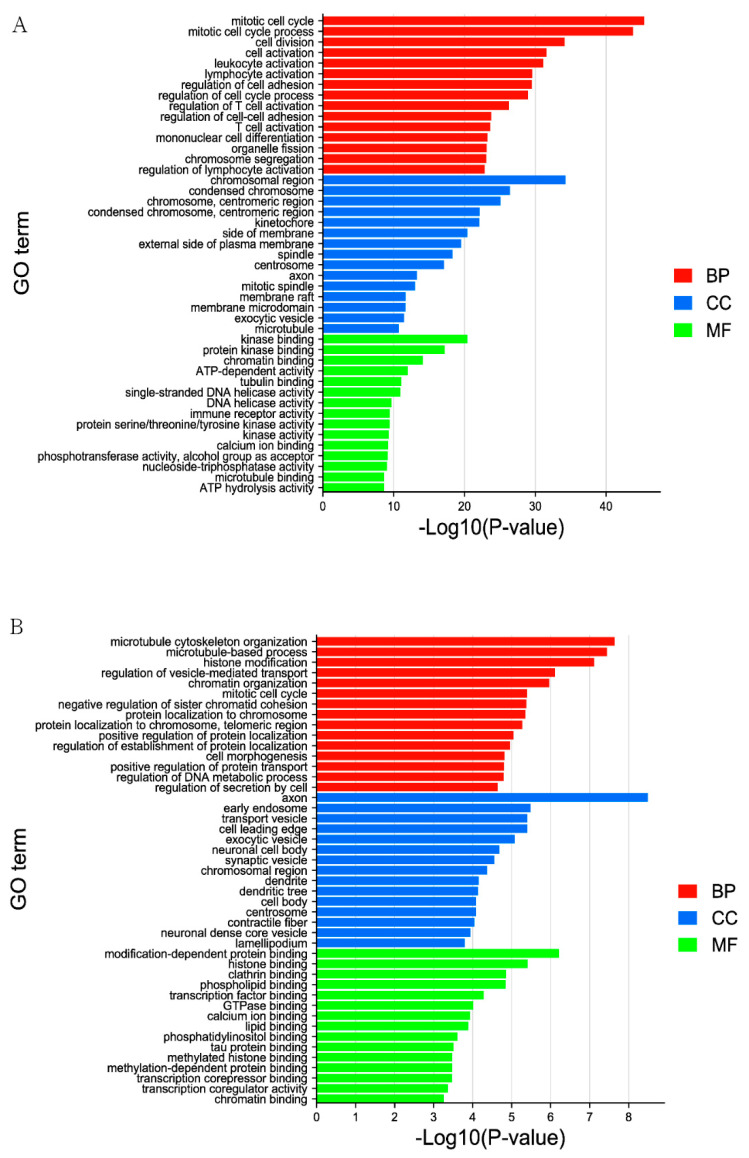
The top 15 enriched GO terms for the identified DELs_targets and DEGs in MM_FT vs. ww_FT. The abscissa and ordinate represent the GO terms and the −Log10 (*p*-value) of enriched genes, respectively. (**A**) GO enrichment terms for DELs_targets in MM_FT vs. ww_FT. (**B**) GO enrichment terms for target genes of DEGs in MM_FT vs. ww_FT.

**Figure 3 genes-13-00849-f003:**
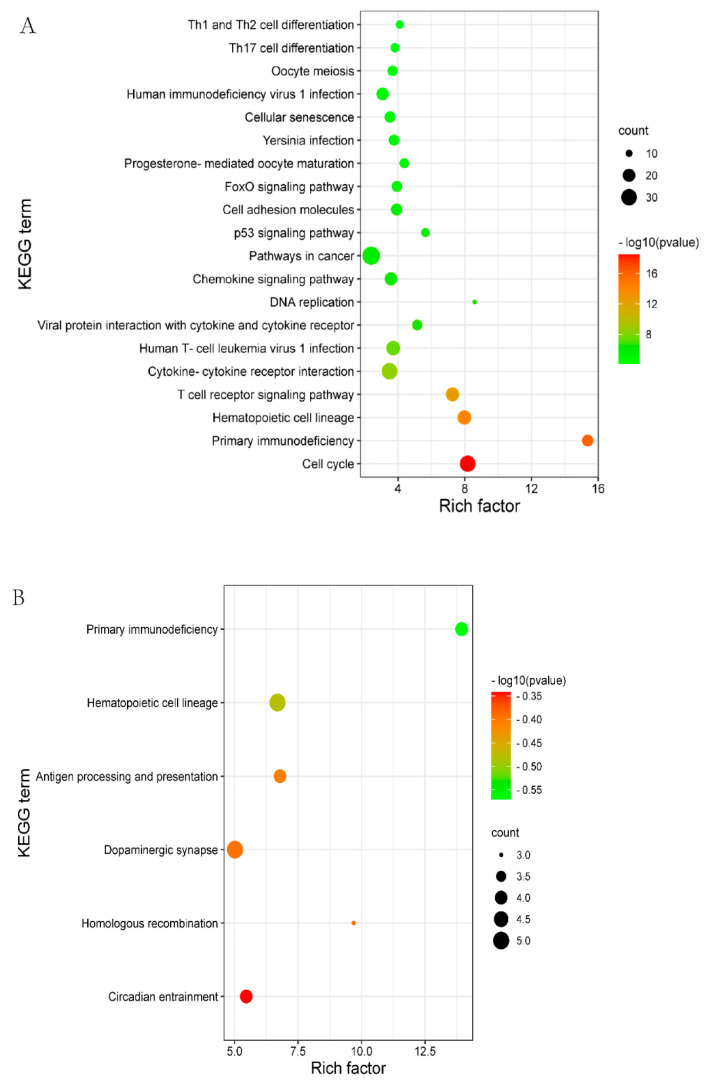
The top 20 enriched KEGG for the identified DEGs and target genes of DELs in MM_FT vs. ww_FT. The abscissa and ordinate represent the number of enriched genes and the KEGG pathways, respectively. (**A**) KEGG enrichment pathways for DELs_targets in MM_FT vs. ww_FT. (**B**) KEGG enrichment pathways for target genes of DEGs in MM_FT vs. ww_FT.

**Figure 4 genes-13-00849-f004:**
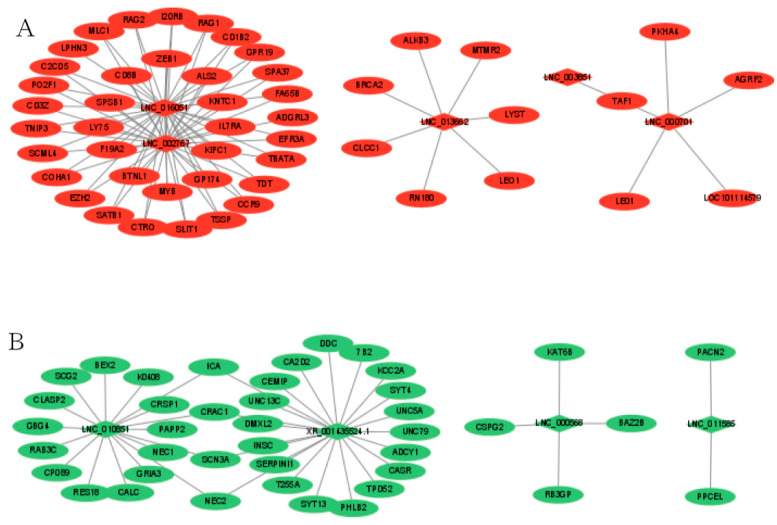
Co-expression of DELs-mRNA after lncRNA targets coincided with DEGs in MM_FT vs. ww_FT, Red (**A**) and green (**B**) indicate up-or down-regulation, respectively.

**Figure 5 genes-13-00849-f005:**
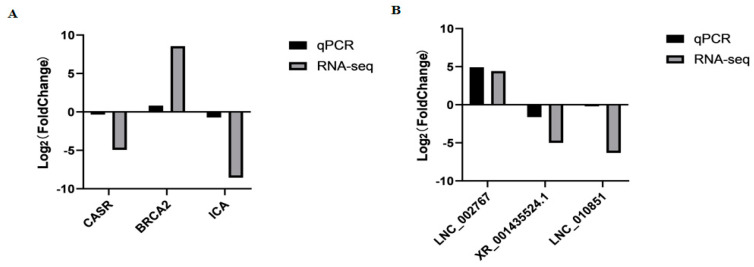
The qPCR validation of mRNAs (**A**) and lncRNAs (**B**) identified by RNA-seq in MM_FT and ww_FT (*p* < 0.001).

**Table 1 genes-13-00849-t001:** Summary of raw reads after quality control and mapping to the reference genome.

Sample Name	Raw Reads	Clean Reads	Clean Bases	Mapped Reads	Mapping Rate (%)	Q30 (%)
MM_F_T_1	97711458	94718122	14.21G	86628641	91.46	95.12
MM_F_T_2	96088926	93452790	14.02G	85202534	91.17	93.78
MM_F_T_3	92644672	90743952	13.61G	83435512	91.95	94.41
ww_F_T_1	90959354	88524974	13.28G	81088654	91.6	94.29
ww_F_T_2	103513712	99977788	15G	90093436	90.11	92.7
ww_F_T_3	84148306	80737376	12.11G	72225658	89.46	92.46

**Table 2 genes-13-00849-t002:** Primer sequences.

Gene Name	Classification	Sequence (5′-3′)
*CASR*	F	TGAGCTTTGACGAGCCTCAG
	R	TGGTCAGGGCGTCATTGTTT
*BRCA2*	F	CTCGACCTGCTTGCTGGTATGC
	R	CGCTGAAGAGTGATGACAAGGGAAG
*ICA*	F	CCGAAGTGTGGATGGCAAGGAAG
	R	ACCCAATGGCGGCATCTGTAAATAG
*LNC_002767*	F	CTGACAGTAGCCTGGTTGGG
	R	ATGACAGGATGTGGGCAGTG
*XR_001435524.1*	F	TAGCTAGGTGGTTCGCTGAC
	R	GAGAGGACGTCTGCAAGGTT
*LNC_010851*	F	GCACTCAGCACACAGGTACT
	R	CCAGAGGAAGACCAACGAGC
*RPL-19*	F	ATCGCCAATGCCAACTC
	R	CCTTTCGCTTACCTATACC

## Data Availability

All data is included in this paper.

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
