# Peer review of "Thyroid Transcriptomic Profiling Reveals the Follicular Phase Differential Regulation of lncRNA and mRNA Related to Prolificacy in Small Tail Han Sheep with Two FecB Genotypes"

_genes, 2022, doi:10.3390/genes13050849_

Round 1

Reviewer 1 Report

The manuscript “Thyroid transcriptomic profiling reveals the follicular phase differential regulation of lncRNA and mRNA related to prolificacy in Small Tail Han sheep with two FecB genotypes” is very well prepared and I think it provides new and valuable information about the regulatory mechanism underlying Small Tail Han sheep prolificacy. The experiments are well designed and the results clearly presented. The experiments are well designed and the results clearly presented by means of images and graphs. The authors identified known and novel lncRNA as well as different expression patterns between the two genotypes. In conclusion, this study provides novel information about the molecular mechanism involved in Small Tail Han sheep prolificacy.

Author Response

We appreciate your careful reading of our paper and the positive comments mentioned above.

Reviewer 2 Report

In this work, the authors aimed to analyze the transcriptomic differences in the thyroid of STH sheep in the follicular phase (FT) between MM group and ww group, detect the DELs and DEGs, and predict their potential function related to sheep reproduction. They found that lncRNAs were implicated in a variety of reproductive functions, such as spermatogenesis, placenta formation, sex hormone response signaling pathways, and gonadogenesis. The thyroid transcriptome analysis revealed differential expression of lncRNAs and mRNAs related to prolificacy in sheep with different FecB genotyping.

In general, the manuscript is well written. Data and discussion of the results are convincing, but some minor inconsistencies need to be clarified. I only have minor corrections and comments to the manuscript, which are outlined below.

Introduction

Lines 42-44: the authors for a better understanding could explain in this paragraph how the three genotypes of FecB are correlated with the litter size of ewes.

Lines 51: which study?

Lines 57-59: add a reference.

Methods

Lines 91-97: add a reference.

Discussion

Lines 271-274: VCAN levels in patients with polycystic ovary syndrome were reduced, demonstrating that VCAN may play a role in ovulatory dysfunction and the pathogenesis of PCOS. In which way?

- I suggest adding at the end of the discussion or in the conclusion a paragraph describing how this data could provide insight into other female mammals.

Author Response

Thank you for your letter and for the reviewers’ comments concerning our manuscript entitled “Thyroid transcriptomic profiling reveals the follicular phase differential regulation of lncRNA and mRNA related to prolificacy in Small Tail Han sheep with two FecB genotypes” (Manuscript ID: genes-1708517). Those comments are all valuable and very helpful for revising and improving our paper, as well as the important guiding significance to our research. We have studied comments carefully and revised them with revision mode in the text. Revised portions are marked in red on the paper. The main corrections in the paper and the responses to the reviewer’s comments are as flowing:

Responds to the reviewer’s comments:

Point 1:

Lines 42-44: the authors for a better understanding could explain in this paragraph how the three genotypes of FecB are correlated with the litter size of ewes.

Response 1:

We thank the raising this question, so we have made a more detailed explanation in this part: On days 9-11 of estrus, the progesterone concentration of FecBBB genotype Booroola Merino sheep was 25% higher than that of FecB++ Merino sheep. The ovulation number of ewes with FecBBB genotype was significantly higher than that of the other two genotypes, FecB mutations inhibit granulosa cell apoptosis in sheep, prevent follicular atresia, promote ovulation and increase litter size. This may be an important physiological mechanism by which the FecB gene affects sheep fertility.

Point 2:

Lines 51: which study?

Response 2:

We are very sorry for our incorrect writing, so we revised it in the text: Hayashizaki et al. shown that in mammals, only 2% of RNA is protein-coding, and more than 70% is the LncRNA.

Point 3:

Lines 57-59: add a reference.

Response 3:

Thank you very much for your question, we have included the appropriate references in the article to demonstrate that lncRNA can associate with some important pathways.

Point 4:

Lines 91-97: add a reference.

Response 4:

Thank you very much for your question. We have added appropriate references to the article.

Point 5: Lines 271-274: VCAN levels in patients with polycystic ovary syndrome were reduced, demonstrating that VCAN may play a role in ovulatory dysfunction and the pathogenesis of PCOS. In which way?

Response 5:

Thanks for raising this important issue, which we have answered in detail: Özler et al. shown that serum VCAN levels in patients with polycystic ovary syndrome (PCOS) were significantly reduced, and there is a positive correlation between VCAN and the expression of ADAMTS-1, indicating that VCAN and ADAMTS-1 may promote each other in the expression process. ADAMTS-1 can affect follicular development, ovulation, luteal formation and degeneration by affecting extracellular matrix (ECM), indicating that VCAN may play a role in ovulatory dysfunction and the pathogenesis of PCOS.

Point 6:

I suggest adding at the end of the discussion or in the conclusion a paragraph describing how this data could provide insight into other female mammals.

Response 6:

Thanks to the reviewer for helpful comments, We added the following words after the discussion: Overall, these five genes play an important role in animal reproduction. In addition, these DEGs are also enriched in important pathways such as cell cycle, p53 signaling pathway and oocyte meiosis, which play a very important role in the biosynthesis of animal reproductive hormones.

We tried our best to improve the manuscript and made some changes in the manuscript. These changes will not influence the content and framework of the paper.

We appreciate for Editors/Reviewers’ warm work earnestly, and hope that the correction will meet with approval.

Once again, thank you very much for your comments and suggestions.
